# Home Dispossession and Commercial Real Estate Dispossession in Tourist Conurbations. Analyzing the Reconfiguration of Displacement Dynamics in Los Cristianos/Las Américas (Tenerife)

**Dennis Hof** 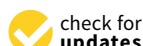

Department of Geography, University of Leipzig, 04103 Leipzig, Germany; dennis.hof@uni-leipzig.de

**Abstract:** Since the onset of the global financial crisis, urban dwellers face an increasing number of obstacles in establishing themselves on the housing market. Against this backdrop, this paper addresses the variegated dynamics of real estate dispossession in the tourist conurbation Los Cristianos/Las Américas on an intra-urban scale. First, I will present the spatio-temporal patterns of dispossession for the period 2001–2015 using the ATLANTE database (CGPJ). Specifically, I analyze mortgage foreclosures and tenant evictions, both for residential and commercial spaces. Second, I delve deeper into local experiences of dispossession of the resident population and their housing and income conditions by means of questionnaires that I conducted in 2018. The data shows that mortgage foreclosures and dispossessions of residential spaces predominate the initial years after the crisis, albeit with varied spatial incidence. However, the increase in tenant evictions from 2014 onwards points to a reconfiguration of displacement dynamics. Indeed, as stated by the interviewees, staggeringly high rent burdens have become the main driver for displacement from both living and working spaces in recent years. Given the ongoing global pandemic, further and more nuanced research is necessary to grasp how these prevailing housing insecurities are shaped during and beyond the coronavirus crisis.

**Keywords:** real estate dispossession; mortgage foreclosure; evictions; short-term rentals; financialization; housing studies; critical geography; Canary Islands

## 1. Introduction

More than a decade since the outbreak of the global financial crisis (GFC), financial insecurity and precarious working conditions continue to shape the quotidian life of many urban dwellers in Spain [1,2]. This is closely linked with the speculative accumulation tendencies of a crisis-prone real estate sector. In recent years, this comprises most notably the rental sector, provoking new dynamics of displacement and dispossession. In addition to the metropolises, these dynamics once again resonate particularly in the insular and Mediterranean tourist regions [3,4].

Starting in the 1960s, the Franco dictatorship certainly paved the way for large-scale real estate-based capital accumulation in Spain, promoting both homeownership and tourism development [5,6]. After the transition, Spain continued to rely on high construction volume and rising real estate prices to foster economic growth: with the country's entry into the European community 1986, foreign investment into the tourism sector further accelerated the Spanish real estate boom [7,8]. At the same time, private household debt rose substantially, given the loose legal regulation of mortgage lending and the prevalence of predatory lending practices and sub-prime mortgages among low-income groups [9]. Following the onset of the GFC and the bursting of the real estate bubble in 2008, construction activity came to a grinding halt, unemployment figures reached record levels, and until 2014, over 570,000 households lost their homes due to mortgage default [1] (p. 322).

On a regional level, this affected mainly the Mediterranean region, as well as the Canary Islands and Balearic Islands, i.e., the hotspots of capital accumulation during the boom years [10–12] (see also Figure 1).

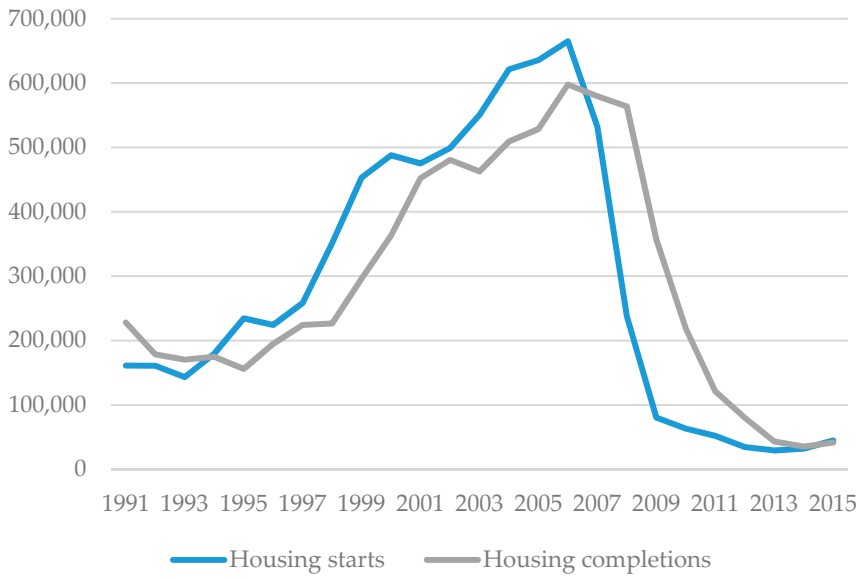

**Figure 1.** Construction activity in Spain (1991–2015). Housing starts and housing completions. Ministerio de Transporte, Movilidad y Agenda Urbana. Own elaboration.

In the aftermath of the crisis, the government-owned clearing bank SAREB (*Sociedad de Gestión de Activos procedentes de la Reestructuración Bancaria*) was responsible for managing and restructuring the plethora of foreclosed properties. In practice, restructuring often involved the large-scale sale of those properties to private-equity firms like Blackstone or Cerberus at discount rates. These global institutional actors then became new players on local real estate markets, inflating the rental sector in form of real estate investment trusts (REITs or SOCIMIs). Coupled with the emergence of very profitable short-term rentals in tourist regions due to disruptive platform providers like Airbnb, this exerted notable pressure upon the remaining affordable long-term rentals. As a result, low-income tenants and shopkeepers struggled to access the rental market. Ironically, after having defaulted on their mortgages or lost their jobs during the crisis, they now depend more than ever on affordable housing in the rental sector. In recent years, though, the number of evictions due to non-payment of rent increased substantially. Despite apparent economic recovery, thus, the pre-pandemic years already pointed towards a new housing crisis, which yet again was politically orchestrated [4,13,14].

Building on these interlinkages, over the last decade, critical scholars have established an extensive field of research regarding the geographies of housing dispossession in Spain. The first studies approached the issue from an interregional scale, analyzing the socio-spatial distribution of mortgage foreclosures and the reasons for the bursting of the Spanish real estate bubble on the basis of the judicial districts [15,16]. The following studies, however, gradually applied the intra-urban scale as a central field of analysis of urban inequalities. Noteworthy in this context are case studies from Catalonia, the Balearic Islands, the Canary Islands, Madrid, and Valencia. While breaking down the intra-urban distribution of dispossessed homes, these studies also analyze the new ownership structures of the previously foreclosed assets and consider the socio-economic contexts of the affected neighborhoods [3,4,13,17–21]. With all these advances being made, difficulties still remain in accessing data on an inframunicipal level. One reason for this is that the existing data of the General Council of the Judiciary (CGPJ) is often defective in terms of providing a comprehensive overview of real estate dispossession. For one part, it does not list commercial and residential real estate separately each. For another part, it lists only

court-ordered foreclosures and evictions. In this sense, the CGPJ only depicts the "tip of the iceberg", as displacement most often involves (semi)voluntary abandonment of the property or cost-neutral out-of-court settlements—in other words, processes, dynamics, and events that researchers struggle to capture [4] (p. 6). Finally, official judicial sources do not give details on the number of members in a household, i.e., with this data it is impossible to know how many people are affected by dispossession (see Parreño Castellano et al. 2019 for a more profound discussion on the limits and merits of CGPJ data) [22].

Following these studies, this paper analyzes mortgage foreclosures and tenant evictions on the intra-urban level of Los Cristianos-Las Américas (LC-LA), a tourist conurbation prominently situated in the south of Tenerife. The overall aim of this paper is to shed light on socio-spatial housing inequalities by (i) exposing how dispossession of residential and working spaces unfolds both temporarily and spatially and (ii) inquiring about the significance of dispossession and displacement in the lives of residents and working people alike. Moreover, I seek to answer to which extent the observed experiences of dispossession resonate with regional and national trends. This also implies exploration of how real estate speculation has shifted towards the rental sector and how this concerns the local population. As mentioned before, several case studies already have dealt with real estate dispossession on the Canary Islands. However, with exception of case studies from the island capitals Las Palmas de Gran Canaria and Santa Cruz de Tenerife, scholars so far have only approached the phenomenon on a regional scale [4,17,23–25]. As these works already hinted at a particular impact on tourist destinations, the case of LC-LA serves as a relevant research area to trace the reconfiguration of dispossession and displacement dynamics on an intra-urban scale. Besides, the municipalities surrounding LC-LA up until the onset of the pandemic comprised the lion's share of Tenerife's short-term rentals, making it all the more worthwhile as a research area. In the context of my research, the underlying database ATLANTE (CGPJ) provides a differentiated consideration of residential and commercial real estate dispossession for the period 2001–2015, thus allowing me to draw conclusions regarding the concernment of working spaces. Looking beyond the mere court-ordered dispossessions, the paper then carries out a supplementary evaluation of questionnaires conducted 2018 in situ. The questionnaires give insights into the experiences of dispossession and their respective socio-economic contexts.

The paper is structured as follows. The next chapter develops the theoretical lens of my investigation, venturing to sketch a rudimentary synthesis of the main insights of critical housing studies regarding dispossession and financialization. Herein, it sketches out emerging trends related to the coronavirus disease 2019 (COVID-19) pandemic, albeit, of course, in a preliminary manner only. The following chapter specifies the empirical data and the methodical procedure, which involves both database analysis and questionnaire evaluation. As a manner of contextual introduction into the research area, the paper then outlines the investment-driven genesis of the tourist settlements in the south of Tenerife. The ensuing main chapter seeks to (i) present the temporalities and spatialities of intra-urban real estate dispossession before (ii) coupling them with the experiences of residents and shopkeepers alike. To conclude, results will be discussed in line with the theoretical and regional lenses. The paper then closes by outlining recommendations for future critical research, also against the backdrop of both (yet lacking) insurgent practices and the ongoing pandemic.

## 2. Housing Financialization and Real Estate Dispossession

Housing has become one of the central social and ecological questions of our time [26,27]. Indeed, critical scholarship has repeatedly sought to raise awareness of an intensifying global housing crisis that, in recent years, comprises inter alia, insecure tenancies, higher entry barriers to local housing markets, soaring rental burdens, and notable increases in displacements and evictions [28–31]. Viewed in a larger context that encompasses decades of neoliberal restructuring, housing dispossession is the consequential concomitant of a wide-scale enclosure of public and urban spheres [32]. More precisely, it is the growing

financialization of the real estate sector that increasingly encloses housing, ranging from mortgage-backed securities and private mortgage-debts to the rise of global landlords in form of stock-listed housing companies or private equity funds [33,34]. The GFC already set a vivid example on how these processes incite displacement and dispossession, most notably seen in traditional homeownership countries like the USA or Spain [35,36]. As ownership rates were consequentially sinking over the last years, those processes now increasingly affect rentals: in form of REITs, where global investors accumulate ever larger portfolios of the rental market, most notably by purchasing (re)commodified social housing stocks and mortgage-defaulted assets, in many cases at discount rates [37,38]. As new landlords, those investors then often push up rents, driving displacements and jeopardizing local living conditions [39]. Coupled with the rise of platform capitalism in form of short-term rentals like Airbnb, those dynamics lead to a further disruption in touristifying neighborhoods [40]. Wachsmuth and Weisler [41] showed how Airbnb creates a new form of rent gap in fashionable and famous neighborhoods, as it requires only minimal capital investment beforehand. As a result, it becomes more appealing for property owners to end long-term rental contracts to shift their focus to short-term rentals. In addition, Airbnb provides professional players with flexible options of buy-to-let investment, given that they can resell tenant-free holiday homes at any time. As a consequence of this Airbnb-driven financialization of the rental sector, global corporate investors increasingly replace local landlords and rely more on affluent tourists than on low-income tenants. This has ramifications for the whole housing market: property prices and rents experience further spikes while urban neighborhoods lose their social cohesion [42–44]. As Hübscher et al. illustrated in the case of the island city Santa Cruz de Tenerife, Airbnb-led touristification increasingly impacts even small and medium-sized cities outside of the traditional tourism markets [45]. This is also related to politically-steered urban branding: in the course of a decades-long tourist marketing and the mercantilization of Spanish cities, city councils actively promoted inner-city touristification. Accordingly, city centers started to resemble more exhibition venues for tourists, lacking space for local businesses, retail, and services for the supply of residents. For city dwellers, the production of tourist spaces is, thus, closely linked to a loss of public spaces and urban life [46,47]. Hence, from the perspective of those affected, displacement and dispossession in all these contexts does not only involve the material loss of residential or commercial spaces but goes hand in hand with the uprooting from everyday references, relationships, and bonds [48].

In light of the ongoing COVID-19 pandemic and its wide-ranging socioeconomic reverberations, it is sheerly inevitable that the pandemic will also affect housing and reshuffle real estate investment strategies. While the pandemic's long-term urban housing outcomes are difficult to predict at this stage, until now, real estate markets seem to have eased some tensions, even in Spain. This is particularly apparent in the higher supply of rentals in previously tense housing markets and tourist destinations: given the (temporary) meltdown of international tourism, many short-term rental providers found themselves forced to head back to the traditional rental market [49,50]. However, many landlords only offer temporary rental contracts as they speculate on a medium-term return to pre-pandemic times. In other words, they want to avoid stable leases that limit their control over their assets at all costs. Therefore, once the pandemic ends and tourism resumes, short-term rentals are bound to return, unless a more ambitious regulatory policy framework is put in place [51–54]. A further explanation for a future revival of short-term rentals relates to stable investor expectations as Airbnb has soared on the stock market this year despite its heavy losses and reliance on institutional lenders [55,56]. Indeed, housing still offers attractive yields at a time of low interest rates, which is why real estate players have so far been reluctant to lower sales prices and rents significantly. Nevertheless, the majority of real estate players also expects a stronger slump in prices for rents and second homes, as well as in construction projects in 2021 and 2022, which, admittedly, is likely to affect real estate agencies with smaller portfolios in particular [57–59]. However, these macro-observations and preliminary projections of real estate sector developments should

not obscure the fact that numerous households—confronted with expiring rent moratoria, job losses, and ongoing economic losses—are already struggling to meet their mortgage or rent obligations [60,61]. What is more, in the midst of a pandemic, landlords and local authorities already enforced numerous (in several cases illegal) evictions in recent months, affecting vulnerable groups in particular [62–66]. After controversial debates, widespread public criticism, and organized protests from tenant movements, the socialist governing coalition has now suspended evictions, at least until May 2021. The eviction moratorium, though, involves financial compensation for house owners for defaulting rents from public funds, regardless if they are private or corporate landlords. In other words, the governing parties once again serve as a political backbone for speculative real estate accumulation. Additionally, the eviction moratorium only provides temporary alleviation while the pending issues to prevent evictions after its expiration and to guarantee affordable housing on the long term remain to be resolved [67–69]. Apart from the housing problems that low-income tenants face, likewise, a notable proportion of people managing local businesses, urban retail, and cultural institutions struggle to keep their businesses afloat, i.e., to sustain their livelihoods [70–73]. In December 2020, the Spanish government introduced an aid package for tourism, gastronomy, and retail in order to remedy the situation. However, the aid package mainly involves rent reductions of up to 50% for the tenants of big landlords; it does not include urgently needed direct financial aid for the shopkeepers themselves [74–77]. All these different modalities of crisis reactions shed light on the political economy of the pandemic: contrary to long-standing mantras of neoliberal economists, it becomes once again apparent that the economy, in this case the real estate sector, is not working entirely independent from the political realm but is intrinsically interwoven with and shaped by the political system. Throughout the last decades, the political system, more than restraining, provided the necessary legal framework for a reckless financialization. Nevertheless, political regulations, depending on their breadth and depth, could fundamentally determine and reshape the course of the economy [78]. Meanwhile, given the absence of deliberate regulatory policies, financial players are already sketching out their future investment plans, expecting to take advantage of faltering prices and defaulted asset sales [79–83]. As for Spain, it is likely that private equity funds yet again will play a dubious lead role in post-pandemic economic restructuring [83]. For instance, in April 2020, Blackstone—the world's largest private equity firm—finished raising a 9.8-billion-euro fund that will specifically target European real estate, allowing it to further consolidate market dominance over Spain's real estate sector [84,85]. At the same time, however, it is also likely that the emerging economic geographies of the pandemic will only exacerbate the austerity-led economic weaknesses of Europe's southern peripheries, given their dependence on tourism and small businesses [86]. Hence, rather than alleviating the housing crisis, the post-pandemic years might actually see a reinforcement of housing financialization and an even more substantial concentration of corporate-financial power in the real estate sector. Real estate will, thus, remain a popular type of investment and form of capital accumulation for transnational investors, thereby further exacerbating the precarious housing and living situations of low-income city dwellers [87].

These inextricable interconnections of housing, financialization, and dispossession, though global in scale, are not monolithic, i.e., they do not unfold uniformly across time and space. While there certainly exist strong linkages between global financial and real estate sectors, urban inequalities and local experiences of dispossession, contrastive considerations of the specificities of national, regional, and local housing policies and housing realities allow for a more nuanced portrayal of the variegated logics and experiences of real estate dispossession [13,88,89]. In this sense, case studies provide a valuable methodical lens to trace the regional or local logics of these globally interdependent dynamics and to understand how they unfold against different contexts. Therefore, the following sections strive to gradually introduce the research area and elaborate their particularities in the context of Spain's real estate development. Before that, however, I will shortly outline the methodical procedure of this paper.

## 3. Materials and Methods

As already mentioned, official data and statistics concerning dispossession in Spain are limited mainly on the regional scale of the judicial districts and do not differentiate between residential and commercial properties. The database ATLANTE (CGPJ), however, is an exception, listing all court-ordered mortgage foreclosures and tenant evictions on the Canary Islands on an intra-urban scale for the period 2001–2015. In addition, it entails nuanced remarks on the assets concerned, allowing for a sophisticated treatment of both residential and commercial properties. However, the judicial data I am dealing with still has some limits that should not be neglected. As Parreño-Castellano et al. [22] point out, a judicial registry naturally lists merely judicial acts or records. While those can be linked in many cases to a property, it is not often possible to associate the records with a person. In other words, I can only assess the number of properties, as the database does not entail information on the number of people affected. In addition, when interpreting the data on mortgage foreclosures and tenant evictions, it is important to bear in mind that loss of housing is not simply the consequence of individual households not being able to pay for rent or for a mortgage loan. In contrast, and as I elaborated in the preceding chapter, it is the material expression of a profound housing crisis and the outcome of a decade-long politically orchestrated enclosure, financialization, and touristification of public and urban spheres in Spain. With all its deficiencies and omittances, the following sections should make clear that judicial data, nonetheless, is a very valuable resource to portray urban inequalities related to real estate dispossession on a geographic microscale.

The methodological framework and methodical procedure draw on these data. As a first step, I aim to identify the temporalities and spatialities of dispossession on an intra-urban scale—divided into mortgage foreclosures and tenant evictions, as well as residential and commercial real estate dispossession. For the preparation of the material, I initially selected all 1695 entries in the database relating to the municipality of Arona in the south of Tenerife. After a thorough data cleansing and examination, 1414 entries remained, of which I georeferenced the 393 entries relating to the research area of LC-LA. Los Cristianos and Playa de Las Américas represent different administrative units, but given their urban morphology, they can be regarded as a single and connected agglomeration. The cartographic handling of the material follows a critical GIS-based (*Geographic information system*) concern to uncover socio-spatial inequalities. In the context of critical GIS, critical cartographers, inter alia, consider the technical possibilities of GIS as a means to enrich critical research [90–92]. As a second step, departing from the temporal-spatial analysis, I will address more recent real estate developments and delve deeper into local experiences of dispossession [93]. According to the principles of qualitative GIS, several scholars highlight the enriching potential of mixed-method approaches for geographical research. In this sense, a combination of quantitative GIS-data with qualitative methods such as non-representative, explorative surveys allows me to further nuance the general tendencies displayed by the database against the context of quotidian life [94,95]. For this purpose, during August 2018, I conducted 50 standardized questionnaires in the research area. In order to reach a greater portion of interviewees with potential experiences of dispossession, I carried out a non-random sample around the spatial hotspots of dispossession that I identified during the database-driven analysis before. On the one hand, the questionnaire aims to acquire knowledge about involuntary displacement of residential and commercial properties and, thus, looks behind the "tip of the iceberg" of court-ordered foreclosures and evictions. On the other hand, it inquires about living and working conditions as well as rent and mortgage burdens in order to evaluate the local experiences of dispossession against the backdrop of contemporary real estate markets. The latter also stems from a long-standing incitement of social polarization in the formation of Tenerife's tourist accumulation regime. As the next section explains, this arises from a specific historic and socio-economic constellation for the organization of capital flows.

## 4. Real Estate Boom in the South of Tenerife

Tenerife's south, with its emblematic tourist hotspot LC-LA, is a prime example of tourism-based real estate growth during Spain's boom years 1998–2007 [96]. In order to gain a more comprehensive picture of the regional dimensions of dispossession and displacement, it is crucial to retrace the genesis of its contemporary accumulation regime, based on tourism and real estate. Before that, though, I will briefly situate the research area in its insular context.

Regional planning policy divides Tenerife into different zones (*Zona Norte*, *Zona Metropolitana*, *Zona Sur*) which themselves are comprised of different municipalities (*municipios*), i.e., local government units. In the southern zone, the most southern municipality of Arona is located. Arona, in turn, is comprised of different population units (*unidades poblacionales*), i.e., distinct cities, towns, places, or villages within a municipality. In Arona, there are 15 population units that can be grouped regarding their distinctive functions in the urban hierarchy [97]. Starting from the higher altitudes, first, the population units north of the highway, Arona, Buzanada, Cabo Blanco, La Camella, Chayofa, and Valle de San Lorenzo, resemble traditional agricultural villages that settled around the more fertile soil in medium altitudes but lost their regional importance throughout the years, given the general reorientation towards coastal tourist areas. Second, Cho, El Fraile, Guargacho, Guaza, and Las Galletas constitute peripheral settlements near the coastal areas but outside the tourist agglomerations. With the exception of Las Galletas, which still is mainly a traditional fisher village, those settlements generally shelter the local population working in the coastal tourist cities. Those are, third, Costa del Silencio, Los Cristianos, Playa de Las Américas, and Palm-Mar. While Palm-Mar is mainly a recent real estate project with holiday homes, Costa del Silencio and LC-LA are bigger agglomerations with ports, shopping centers, hotels, shops, bars, restaurants, and other vital facilities of the tourist industry. Already, this short overview of the urban hierarchy of Arona should give an idea about the drastic shifts from agriculture towards tourist real estate that the economy of the archipelago experienced in a relatively short period of time.

For many decades now, the Canary Islands have been an important pillar of Spanish real estate growth. During the boom years, increases in construction volume and property prices significantly exceeded national dynamics [17] (p. 27). This stems largely from a territorial reorganization of the archipelago since the 1960s, in which the political elites promoted new tourist accumulation regimes, especially in the south of Tenerife. Once an arid region, the introduction of agricultural irrigation systems modernized the island's southern municipalities in the 1950s. Given widespread expectations of higher profitability in tourist destinations, water supply then gradually shifted from agricultural to tourist land use on the coastal areas. This meant, in other words, "the exploitation of spaces which had been hitherto undervalued" [98] (p. 30). New investments in fixed capital were then necessary as these new economic activities "required the creation of a physical infrastructure adequate to the needs of production and circulation of capital" [98] (p. 37). To this end, large local landowners cooperated closely with municipal administrations, which steered and promoted tourist real estate projects on the coastal areas and legally secured the planning and implementation of these projects via new zoning regulations. Particularly the *Plan Insular de Tenerife* (1969), commissioned to the planning agency Doxiadis Ibérica, set the tone for the following decades; it placed all the major upcoming infrastructure projects (autopista del Sur, aeropuerto Reina Sofia) in the south of Tenerife and designated Los Cristianos in the municipality of Arona as the future tourist center of the island [99]. Until 1980, real estate actors opened a total of 2967 hectares of land for tourism—1099 hectares in Arona alone. This close collaboration of owners and the municipal agents illustrates how the production of tourist spaces constitutes a "spatial fix" of capital in the built environment in which the state plays a key role in terms of its adapted regulations, its planning practices, and its ideological backing of capital interests [100]. Tragically, however, for many local landowners, while the initiatives for capital-switching came from local capital, in many cases, only joint ventures between big landowners from the Canary Islands

and peninsular or foreign investors proved to be successful. Given their lack of capital, smaller local landowners often could not establish themselves as tourist developers, forcing them to sell their plots to their wealthier competitors. Worth mentioning in particular is the joint venture between Antonio Domínguez Alfonso and the Catalan entrepreneur Rafael Puig Lluvina, who implemented the 281-hectare project Playa de Las Américas Fase III, thereby linking Los Cristianos morphologically with the previously completed Fase I and Fase II. In a nutshell, in less than three decades, this alliance of urban planning, large landowners and foreign investors managed to fundamentally redirect the accumulation regime from agricultural yield to tourist and real estate yield. As García-Herrera puts it, urbanization in the Canary Islands "took place without industrialization [98] (p. 36). This led to the configuration of LC-LA as Tenerife's tourist epicenter, determining—besides the metropolitan area Santa Cruz/La Laguna—the economic and spatial orientation of the island [99,101]. As a side effect of this fast reorientation of the accumulation regime, however, severe structural deficiencies in the urban process became apparent: housing shortage was very common among low-income families who, prior to the 1990s, needed to rely on self-help housing [98]. This type of housing then became the dominant morphology that constituted the new emerging peripheral settlements near the coastal areas. In the boom years of the 1990s, Spain's entry into the EU led to a further intensification of real estate investment, with German capital in particular participating in new tourism projects. Most recently, real estate development comprised the rest of the island, especially including peripheral settlements [102]. Despite all this, locals and non-EU migrants working in tourism and construction apparently have not been involved substantially in the profits of real estate growth. On the contrary, employment in these industries, which usually required no or very low qualifications, was highly precarious and characterized by temporary employment and low income [103]. In addition, social benefits and social housing policies were weak in the Canary Islands, even by national standards, so that raising a mortgage was often the only viable option to access housing [17,23]. In this sense, private mortgage debt took a firm hold in the quotidian life of the population [1] and further fueled social polarization.

Thus, the bursting of the real estate bubble only exacerbated pre-existing structural precarities and vulnerabilities. The construction volume stagnated from 28,798 newly initiated construction projects in 2006 to only 394 in 2014; at the same time, unemployment and insolvency erupted [17] (p. 29). On an interregional level, the subsequent waves of foreclosures affected the social fabric of the Canaries archipelago most severely, alongside Catalonia: in the period 2007–2012, on average, 8.8 foreclosures per 1000 inhabitants materialized [104] (p. 6). On an intraregional level, the tourist islands Fuerteventura and Lanzarote on a whole and the tourist regions in the South of Gran Canaria and Tenerife proved to be particularly affected [23]. One can get a first glimpse of the scope and magnitude of these events in the south of Tenerife by examining the variations in population growth in the municipality of Arona—given the absence of real estate-related indicators on an infra-municipal level (see Table 1). In my interpretation it is fair to assume that the very high population growth until 2008 accounted for excessive real estate growth during the boom years, while the declining and partly negative trends from 2008 onwards—especially in the coastal conurbations like Costa del Silencio and Los Cristianos—pointed to crisis-related housing losses.

Contrary to a widespread perception of real estate recovery in recent years, low-income residents are still struggling to gain a foothold in the housing market. On the one hand, as a result of defaulted assets and restrictions in mortgages, they depend more than ever on the rental sector. On the other hand, the penetration of the rental market with global investment funds and short-term rentals has led to significant rent increases, also in the Canary Islands. In this context, tenant evictions are on the rise since the outbreak of the housing crisis, as the cases of Las Palmas de Gran Canaria and Santa Cruz de Tenerife illustrate [4,24]. For urban agglomerations outside the island capitals, there are no conclusive studies in respect thereof yet. However, the massive concentration of Airbnb

offers in the hands of a few professional hosts in Arona and Adeje indicates that recent displacement dynamics are again affecting mainly the tourist destinations [105]. Especially in Los Cristianos, short-term rentals seem to pile up outside the hotel strongholds, i.e., in the residential areas [106] (p. 24). In the following empirical sections of this paper, I will explore these linkages in more detail.

**Table 1.** Population growth in % in the municipality Arona (2000–2017).

| Population Unit | 2000–2008 | 2008–2017 | 2000–2017 |
|---|---|---|---|
| Municipality Arona | 97.58 | 3.99 | 105.46 |
| Arona | 29.52 | −5.08 | 22.93 |
| Buzanada | 85.99 | 9.94 | 104.48 |
| Cabo Blanco | 49.80 | 10.41 | 65.39 |
| La Camella | 60.52 | 11.32 | 78.70 |
| Los Cristianos | 85.10 | −9.03 | 68.38 |
| Cho | 262.62 | 32.69 | 381.17 |
| El Fraile | 90.52 | 20.47 | 129.52 |
| Las Galletas | 118.76 | 5.67 | 131.16 |
| Guaza | 133.88 | 12.24 | 162.50 |
| Costa del Silencio | 242.59 | −12.39 | 200.14 |
| Chayofa | 167.53 | 0.40 | 168.60 |
| Palm-Mar | 533.89 | 91.15 | 1111.67 |
| Playa de Las Américas | 91.09 | 1.32 | 93.60 |
| Valle de San Lorenzo | 71.39 | 3.76 | 77.84 |
| Guargacho | 127.13 | 1.68 | 130.94 [1] |

[1] INE. Nomenclátor: Población del Padrón Continuo por Unidad Poblacional. Own elaboration.

## 5. Dispossession and Displacement in Los Cristianos/Las Américas

After having outlined the theoretical and regional contexts, the main section of this paper now proceeds with a nuanced analysis of dispossession and displacement dynamics in LC-LA. In a first subsection, I will expose the temporal and spatial distributions of court-ordered dispossessions on an intra-urban level. After that, against the context of a real estate market penetrated by price-driving short-term rentals, in a second subsection, I will look at more recent developments and local experiences of dispossession.

### 5.1. Temporalities and Spatialities of Dispossession

Figure 2 provides an insight into the temporal evolution of dispossessions. In line with national trends, the major share of dispossessions in Arona coincides with the years following the outbreak of the housing crisis. Accordingly, this share mainly involves mortgage foreclosures. For the period during the boom years, the database does not display any mortgage foreclosures in the research area. Even on the municipal level, only seven mortgage foreclosures are recorded in that period, all between 2006–2008. This near absence in the pre-crisis years again underscores how mortgage foreclosures only achieved widespread, albeit doubtful, fame through the housing crisis. However, the last two years of the observation period reveal a slight but notable increase in tenant evictions. Table 2, in turn, elaborates these tendencies on the infra-municipal level. Mortgage foreclosures still dominate in almost all population units of the municipality, yet Los Cristianos and Playa de Las Américas record the highest percentage of tenant evictions. On top of that, the conurbation also concentrates the biggest number of dispossessions: 393 court-ordered proceedings, which account for 27.8% of all dispossessions recorded in Arona. If we relate the figures with the population, it becomes further apparent that, apart from the peripheral settlements of Guaza (29.68‰) and Guargacho (37‰), the dispossession rates are highest in Playa de Las Américas (23.7‰). Therefore, the conurbation deserves special consideration with regard to both the amount of dispossessions and the degree to which they affect the population.

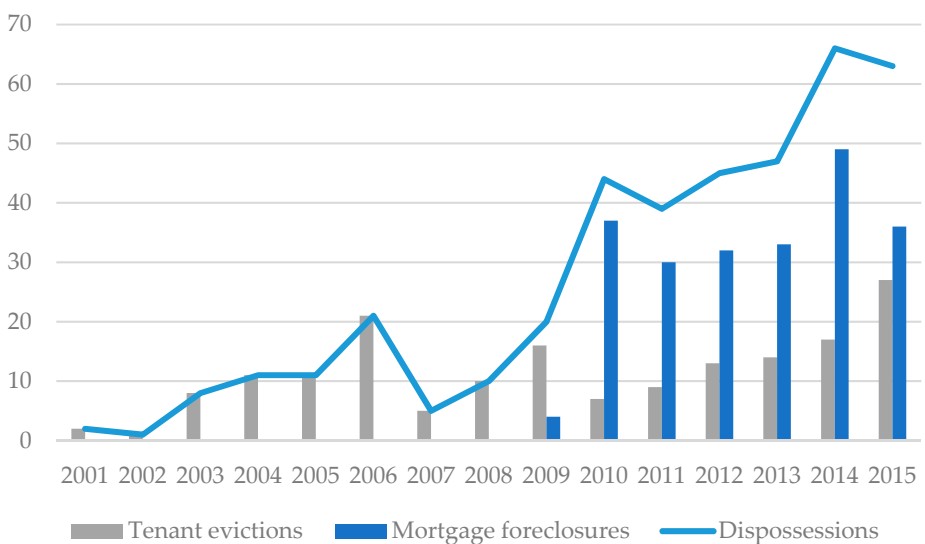

**Figure 2.** Evolution of dispossession in Arona (2001–2015). Database ATLANTE. Own elaboration.

**Table 2.** Real estate dispossession in the population units of Arona 2001–2015.

| Population Unit | Dispossession | Tenant Evictions (%) | Mortgage Foreclosures (%) |
|---|---|---|---|
| Municipality Arona | 1414 | 30.55 | 69.45 |
| Arona | 34 | 11.76 | 88.24 |
| Buzanada | 50 | 22.00 | 78.00 |
| Cabo Blanco | 100 | 17.00 | 83.00 |
| La Camella | 69 | 17.39 | 82.61 |
| Los Cristianos | 272 | 40.07 | 59.93 |
| Cho | 55 | 29.09 | 70.91 |
| El Fraile | 121 | 33.88 | 66.12 |
| Las Galletas | 79 | 32.91 | 67.09 |
| Guaza | 59 | 27.12 | 72.88 |
| Costa del Silencio | 182 | 27.47 | 72.53 |
| Chayofa | 34 | 26.47 | 73.53 |
| Palm-Mar | 28 | 25.00 | 75.00 |
| Playa de Las Américas | 121 | 52.07 | 47.93 |
| Valle de San Lorenzo | 136 | 19.12 | 80.88 |
| Guargacho | 74 | 33.78 | 66.22 [1] |

[1] Database ATLANTE (CGPJ). Own elaboration.

Indeed, already the temporalities in LC/LA exhibit peculiarities that distinguish it from the municipal level. Although mortgage foreclosures also dominated the years following the outbreak of the crisis (49 cases in 2014 alone) and throughout the whole observation period (56.23%), tenant evictions bear a much stronger role than in other population units of Arona. Tenant evictions in the pre-crisis years 2001–2007 alone accounted for 15.01% of all dispossessions in the research area. On the municipal level, the corresponding figure amounted to only 9.33%. This illustrates that unstable and insecure housing conditions in the rental sector already attained significance in the pre-crisis years [107]. The steady rise in tenant evictions from 2010 onwards in turn indicates that the dynamics of displacement and dispossession increasingly shifted to the rental sector in recent years [4].

However, it is also clear that these dynamics do not affect the entire urban area equally. Based on the findings that the coastal cities in Arona particularly registered negative population trends from 2008 onwards, an initial analysis of the census sections located in LC/LA found that especially the sections in central locations around the promenades and Montaña Chayofita displayed the greatest population losses since 2008. As a more nuanced point density analysis of all the georeferenced dispossessions indicates, this also

reflects the spatially dominant patterns of intra-urban real estate losses (see Figure 3). In that regard, the historical center of Los Cristianos recorded the highest density of dispossessions, essentially being mortgage foreclosures. Tenant evictions, by contrast, did not cluster in a similarly prominent way. While they predominated generally in Playa de Las Américas (52.07% of all tenant evictions), they were much more scattered (not taking into account the urbanización El Camisón), owing to the more expansive morphology of this segment of the conurbation. These patterns reflect case studies from the Balearic Islands, where researchers also observed clusters of mortgage foreclosures and scattering of tenant evictions [3,108]. Another inner-city hotspot worth mentioning is the Edificio Edén located northeast of the historical center of Los Cristianos. A temporary employment agency and the municipality's employment agency operate in the building, which, thus, serves as a reference point for residents working in the local tourism and services industry. The concentrations of dispossessions recorded there, therefore, offer initial insights into the socioeconomic repercussions of dispossession, which I will discuss in more detail in the following chapter.

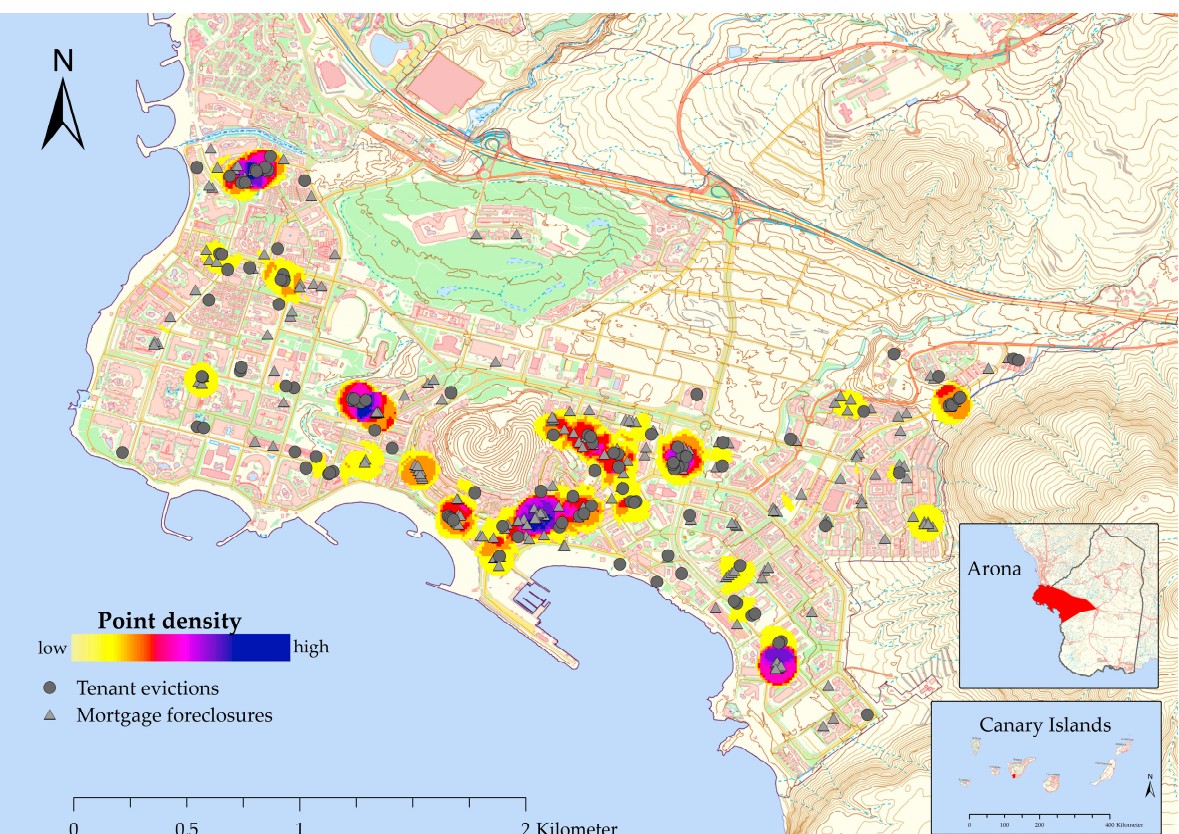

**Figure 3.** Density of real estate dispossession in Los Cristianos/Las Américas (2001–2015). Database ATLANTE (CGPJ). Own elaboration. Mapa topográfico Grafcan (2014).

As mentioned before, a distinctive feature of the database is that it allows me to differentiate between residential and commercial properties (stores, restaurants, offices). In that manner, I could assign 70 of the 393 georeferenced dispossessions to commercial properties that cluster mainly around shopping centers in Playa de Las Américas; the northwesternmost foothills exhibit the highest concentration. Hence, the repercussions of the housing crisis proved to not only jeopardize residential but also working spaces.

In the following subsection, I will examine these interrelationships in more detail with regard to the socio-economic situations and housing conditions of the interviewees. In this way, I seek to develop a more qualitative and differentiated approach to capture the multifarious ways and urban contexts in which dispossession affects people.

### 5.2. Local Experiences of Dispossession and Displacement

The intra-urban analysis of the temporalities and spatialities of dispossession patterns already brought to light that tenant evictions have been increasing in recent years and that real estate dispossession also affects commercial properties to a considerable extent. Now portraying the key findings of the standardized questionnaires conducted in the research area in 2018, I aim to further shed light on the concomitant precarizations in the access to living and working spaces. For this purpose, I will (i) outline the social profile of the interviewees before (ii) identifying contemporary and dominant forms of dispossession, also in terms of their spatial context. The section closes (iii) with an account of present housing uncertainties against the backdrop of high housing cost burdens.

Most of the interviewees were 18–50-year-old employees or self-employed shop keepers with comparatively high educational qualifications. Scholars have already highlighted the high level of over-qualification in the tourism sector, i.e., that job offers and income opportunities do not necessarily match the degrees achieved [103,109]. In addition, a large share of 65% of the interviewees did not live in the conurbation for more than four years, which is only consequential, given the predominance of migration backgrounds (mainly South America and EU member states). Accordingly, the local experiences of dispossession reconstructed here largely reflect more recent years and are, amongst other factors, shaped by alien status.

Considering the housing situation revealed that a large part of the interviewees now lived in rental housing, while mortgages did not attach much importance anymore. Thus, it did not come as a surprise that the interviewees saw rent increases as the essential reason for having to abandon and leave their living and working spaces involuntarily (see Figure 4); indeed, of the 50 respondents, 62% reported experiencing involuntary loss of residential and commercial properties themselves or hearing it from others. This is also true for people with migration backgrounds; of the 29 interviewees with foreign origin (outside of Spain), 55% experienced real estate dispossession, either directly or in their social surroundings. While the data is far from being conclusive here, it clearly is in line with a more recent case study from Las Palmas de Gran Canaria that shed light on the unequal exposure of the foreign population to real estate dispossession [110]. At the same time, the lower purchasing power of customers as a result of the crisis proved to be a major problem for shop keepers in particular, forcing them to close their stores. Since I conducted the majority of the interviews with shopkeepers, consequently two-thirds of the experiences of dispossession related to commercial properties. The involuntarily abandoned stores were, as might be expected, locally operated, tourism-oriented bars and restaurants, as well as clothing and decorating stores. Today, many of these stores are vacant, but the majority are managed by new owners, for whom some of the interviewed former shopkeepers now work as employees. By contrast, job losses and mortgage debt have ceased to be relevant drivers for displacement. An interviewed real estate agent confirmed this general shift, stating that since at least 2015, rent increases have started to be the dominant drivers for displacement, especially with the advent of short-term rental platforms such as Airbnb. That being said, there seems to be little sign of the housing vacancy that became such a common feature of the housing crisis as 36% of respondents had no knowledge of vacant apartments, while 52% stated that not a single apartment in their immediate area was vacant. In turn, 36% knew about short-term rentals in their neighborhood with individual respondents stating that, today, most apartments were rented to more affluent tourists. In other words, in the context of the financialization of the rental sector, new dynamics of dispossession and displacement are unfolding at the national [14] and regional level [4].

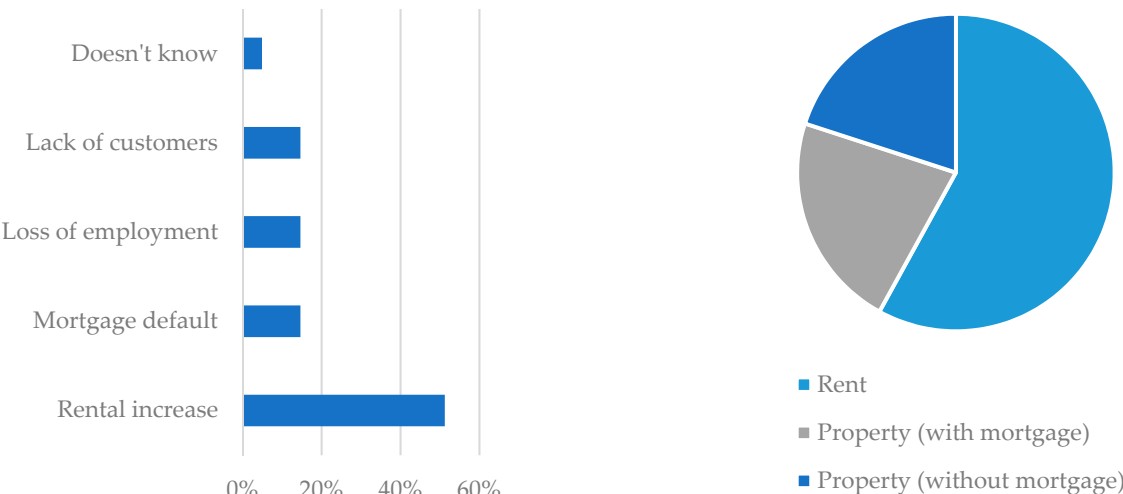

**Figure 4.** Reasons for dispossession (**left**) and type of ownership (**right**). Own elaboration.

The interviewees related their experiences of dispossession mostly to Los Cristianos and Playa de Las Américas, in part also to Tenerife and Lanzarote, i.e., to nearby local and regional contexts. However, many interviewees were unaware of the whereabouts of the displaced persons (insofar as they themselves were not affected)—which makes it clear that dispossession often goes hand in hand with the loss of everyday references, relationships, and bonds [48]. More general statements, in turn, argued that most of the migrants concerned had returned to their countries of origin, while many locals were forced to return to live with their families.

Furthermore, more than a decade after the GFC, a fresh start on the housing market still is out of the question, as the current income levels of the interviewees revealed staggeringly high mortgage and rent burdens. Indeed, out of the 80% of interviewees who lived in rental or mortgaged housing, 47.5% had to dedicate more than 40% of their income to monthly housing costs. These figures clearly exceed the threshold of 30%, which scholars frequently invoke in the debate on affordable housing [111]. On top of that, housing cost burdens are closely related to experiences of dispossession; those who have to bear high monthly housing costs tend to know more about dispossession and vice versa. In this sense, dispossession and displacement unfold especially in the quotidian life of low-income groups. Hence, more than ten years after the outbreak of the housing crisis, housing insecurity still shapes contemporary urbanism and the fortunes and misfortunes of urban dwellers in LC-LA.

## 6. Discussion

The interconnections of housing financialization and dispossession, observed at the global level, bear their own manifestations in the Canary Islands. The case study presented here pointed out that dispossession and displacement are increasingly reverberating in the rental sector in recent years—in a similar fashion to what scholars have already been observing at the global [30] and national level [14]. At the same time, however, the case study emphasized the contextual nuances or variegated logics of these global processes [89]. In Spain, housing insecurity stems in large part from politically questionable practices—involving inter alia, illegal mortgage lending, and the creation of REITs—that have enabled real estate speculation for decades without introducing any significant measures to guarantee affordable and accessible housing for low-income groups [13]. In the Canary Islands, this is closely linked with the formation of tourist accumulation regimes, set in motion by an alliance of big local landowners, foreign investors, and regional administrations from the 1960s onwards and backed by precarious labor and housing conditions [98,99]. This social polarization set the stage for the socially devastating effects

of the housing crisis and its persistence until today. It is important to note that the state played a key role here. With its adapted regulations, planning practices, and the ideological backing of capital interests, it enabled and steered the production of tourist spaces as a "spatial fix" for capital [100].

The case study of LC/LA allowed me to pin down this reconfiguration of displacement and dispossession dynamics and their unfoldment at the intra-urban level. The major share of dispossessions are mortgage foreclosures, which mainly concern residential properties and the historical center of Los Cristianos. At the same time, however, a notable proportion of dispossessions also involves commercial properties. Additionally, tenant evictions have experienced an upswing in recent years. In this regard, the standardized questionnaires illustrated that the recent financialization of the rental sector [14] in practice is tantamount to staggeringly high housing cost burdens. Rising rents, in this sense, ostensibly displace low-income households and shopkeepers. Considering the genesis of Tenerife's accumulation regime, it is striking how low-income groups and vulnerable households for decades now have to bear the burden of an ideological project that (i) rests on the assumption of endless tourist valorization of real estate and was and is foremost (ii) led by capital interests but ultimately (iii) backed and fueled by state institutions. The theoretical contribution of this study, thus, lies in linking these distinct pathways of capital-switching in the Canary Islands since the 1950s with contemporary global debates on financialization and touristification. In this sense, it provides a more nuanced framework to trace and assess the emerging and prevailing urban inequalities of the archipelago.

Further research and data are necessary to gain more comprehensive insight into these dynamics. This implies outlining the intra-urban court-ordered dispossessions subsequent to the observation period, which should be spatially compared with the available supply of short-term rentals and their development over time. Along with this, in-depth qualitative interviews would allow me to delve deeper into the narratives and even lived experiences of dispossession beyond the mere quantitative data available, taking into account the perspectives of real estate agents, urban planners, social workers, and those affected by dispossession. Beyond that, expanding the research on intra-urban ownership structures would provide crucial and possible insights into the penetration of institutional investors into urban housing markets after the repossession of assets. On a conceptual level, the relationship between the price developments of local urban housing markets and the dispossession of low-income earners also requires closer consideration. This might imply working with extensive surveys to expand on the CPGJ-data's limits regarding household members.

However, more comprehensive social housing policies and a political commitment to implement them are also needed. In the Canary Islands, critical scholars debated and argued for a stronger regulation of the speculative tourist sector for years. Previous efforts to introduce a moratorium on short-term rentals have fizzled out though [4,112]. Given that national and regional policies, thus, show little willingness to provide affordable housing, the hope of critical scholarship rests on the insurgent practices of social movements [113]. In the years following the outbreak of the housing crisis, however, unlike in other Spanish cities, there has been no significant mass movement against housing displacement in the Canary Islands [114]. Regarding the south of Tenerife, the dispersed distribution of peripheral working-class settlements, combined with a migrant population working in the tourism sector but only loosely rooted locally, is likely limiting the possibilities for social mobilization. Future research, therefore, should also aim to engage residents and those affected more actively in the research process, as this has proven to be a helpful means of promoting collective self-determination [115]. Critical geographic research, thus, has a central role to play in mobilizing the numerous people affected by the displacement of living and working space in the Canary Islands, who have not yet achieved a meaningful and resistant form of collective organization.

What is more, the novelty of the current situation underscores the necessity of continuous critical research as it is vital to grasp how housing insecurities are shaped during and

beyond the COVID-19 pandemic. At a first and general glance, rather than being a social leveler, the ongoing pandemic proves to exacerbate pre-existing inequalities [116]. While the pandemic's more specific consequences in terms of its urban housing outcomes are difficult to predict at this stage, it is very likely that the relentless dynamics of real estate financialization will make the housing and living conditions of low-income city dwellers more precarious [49–87]. Tragically, as the pandemic advances, protest options are further hampered, because collective organizing, such as rent strikes, can only take place online or in compliance with social distancing [117]. Meanwhile, global investors like Blackstone are already preparing for a novel speculative round of housing accumulation on a large scale [79–86]. Precisely because of these uncertainties and ambiguities associated with the pandemic, it will be more necessary than ever for critical research to focus and expand on the dynamics of accumulation and dispossession. In that regard, the potential of critical housing research resides in providing more comprehensive and transferable knowledge of the pandemic's housing consequences and the obstacles that urban dwellers will face beyond the 2020s. To conclude with a hopeful notion, the acquired critical knowledge—if made properly available for the public in general and the people affected in particular—can then serve as a catalyst for societal change to expand (again) the social function of housing.

**Funding:** This research received no external funding.

**Informed Consent Statement:** Since primary data was collected by means of questionnaires, all subjects involved in the study were informed about their aims and publication of the results.

**Data Availability Statement:** The Consejo General del Poder Judicial (CGPJ) provided the AT-LANTE database to colleagues of the Universidad de La Laguna. Unfortunately, the database is not avalaible online.

**Acknowledgments:** This paper has been possible thanks to the research projects of the Ministerio de Ciencia, Innovación y Universidades. Agencia estatal de investigación: El conflicto urbano en los espacios de reproducción. La vivienda como escenario de conflictividad social (RTI2018-094142-B-C22); from the Ministerio de Economía y Competitividad. Agencia estatal de investigación, y Fondo Europeo de Desarrollo Regional (FEDER): Crisis y vulnerabilidad social en ciudades insulares españolas; the project Contested Territories financed by the European Union research and innovation program Horizon 2020 with the reference Marie Skłodowska-Curie number 873082 (CON-TESTED_TERRITORIES); as well as the Pre-Doc Award program at Leipzig University. I further acknowledge support from Leipzig University for Open Access Publishing.

**Conflicts of Interest:** The author declares no conflict of interest.

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
