# Peer review of "Home Dispossession and Commercial Real Estate Dispossession in Tourist Conurbations. Analyzing the Reconfiguration of Displacement Dynamics in Los Cristianos/Las Américas (Tenerife)"

_urbansci, doi:10.3390/urbansci5010030_

Round 1

Reviewer 1 Report

This is 80 percent of an excellent, concise analysis of the aftermath of the Global Financial Crisis, and the waves of foreclosures and dispossessions that have displaced the 'moral hazards' of capital onto the daily struggles of families and households.  The conceptualization, literature review, and methodological framing of the study are impressive.  There are multiple valuable theoretical syntheses of the literatures on these topics, great connections with how platform dynamics such as AirBnB are altering rent gap dynamics -- and all of this is nicely adapted to the distinctive contexts of tourist-driven urban growth machines such as Tenerife.  Fascinating planning and development histories (another landscape shaped by Doxiadis!).  My main concern is section 5.2, "Lived experience," to the end of the paper.  The results here are empirically brief.  There's just too few empirical results here to do justice to the very complex, qualitative facets signaled by the phrase "lived experience."  And so the discussion and conclusions are therefore quite limited.  If there's more empirical information available from the questionnaires, use it to sharpen the overall contribution of the study.  Even if you don't add in more empirical information, you could reframe the first three paragraphs of the discussion section, and scale back the final two paragraphs.  That would make space for a revised discussion that emphasizes what is theoretically unique about the findings.  This could be more than just a case study of financialization and the aftermath of crisis.  It could be a way of analyzing how Harvey-Lefebvre circuits-of-capital dynamics are being fundamentally transformed in urban spaces that are built upon the assumption of an endless Castells "space of flows" of tourist valorization of real estate -- and how state institutions subsequently force the risks of the bubble onto vulnerable households and families when the inflation of fictitious capital eventually collapses.

As I noted, I think this is a good paper, and it's 80 percent of the way to an excellent contribution. 

Reviewer 2 Report

Dear author,

This paper is well written for the purpose of research. In the future, I hope you will continue to conduct comparative analysis studies on commercial real estate dispossession caused by COVID-19.

Author Response

This paper is well written for the purpose of research. In the future, I hope you will continue to conduct comparative analysis studies on commercial real estate dispossession caused by COVID-19.

Thank you for your feedback! I am currently working to prepare my PhD on Post-Covid-19 Housing Financialization, though with a regional focus on medium-sized German metropolises. That being said, I wish to expand my research to a comparative European scope during my Post-Doc phase.

Reviewer 3 Report

It is an interesting work that has three strengths in relation to the existing literature: first, it focuses the analysis on an insular tourist city in the fourth European tourist periphery, something that has been little or no analyzed; second, it attempts to contextualize the phenomenon of dispossession in the recent scenarios constituted by the development of vacation homes, the internationalization of the real estate market and the current pandemic. Finally, it combines quantitative analysis with qualitative surveys.

Since the article works with judicial sources, it deserves a somewhat deeper reflection on its characteristics. You can find references in Parreño Castellano, J. M., et al. (2019). Real estate dispossession and evictions in Spain: a theoretical geographical approach. BAGE, (80), 11.

Also in the context of the Canary Islands, the relationship between foreclosure and migration might to be taken into account, since in the tourist areas of the islands quite possibly, as the author suggests, a part of the people who suffer dispossession are of foreign origin.

Parreño Castellano, J. M., Domínguez-Mujica, J., & Moreno-Medina, C. (2020). Real estate dispossession, income and immigration in Las Palmas de Gran Canaria (Spain). BAGE, (87).

In the introduction, two statements must be clarified:

- Line 40. "given the widespread practice of illegal mortgage ..."

- Line 43. Until 2014, 570,000 households lost their homes… Source of this statement?

In Figure 2 the mortgage foreclosures between 2001 and 2008 are not represented. If it is a lack in data, it must be indicated and explained. It is not correct to think that in that period there were no foreclosures in the analyzed area, as suggested in line 364.

In the context of the Canary Islands, two recent references are: 

García-Hernández, J. S., Armas-Díaz, A., & del Carmen Díaz-Rodríguez, M. (2020). Desposesión de vivienda y turistificación en Santa Cruz de Tenerife (Canarias-España): los desahucios a inquilinos en el barrio de El Toscal. Boletín de la Asociación de Geógrafos Españoles, (87). 

In this paper, the authors have used qualitative methods. The work might be useful since a comparative point of view. 

The development of vacation rentals in island cities is studied in 

Huebscher, M., Schulze, J., zur Lage, F., & Ringel, J. (2020). THE IMPACT OF AIRBNB ON A NON-TOURISTIC CITY. A CASE STUDY OF SHORT-TERM RENTALS IN SANTA CRUZ DE TENERIFE (SPAIN). ERDKUNDE74(3), 191-204. 

Finally, a brief presentation of the study area is necessary. Population entities are mentioned and for the international reader it is difficult to make a territorial reading without prior information.
